# WATCH OUT!! YOUR CONFIDENCE MIGHT BE A REASON FOR VULNERABILITY

## ABSTRACT

The tremendous success of deep neural networks (DNNs) in solving 'any' complex computer vision task leaves no stone unturned for their deployment in the physical world. However, the concerns arise when natural adversarial corruptions might perturb the physical world in unconstrained images. It is widely known that these corruptions are inherently present in the environment and can fool DNNs. While the literature aims to provide safety to DNNs against these natural corruptions they have developed two forms of defenses: (i) detection of corrupted images and (ii) mitigation of corruptions. So far, very little work has been done to understand the reason behind the vulnerabilities of DNNs against such corruption. We assert that network confidence is an essential component and ask whether the higher it is, the better the decision of a network is or not. Moreover, we ask the question of whether this confidence itself is a reason for their vulnerability against corruption. We extensively study the correlation between the confidence of a model and its robustness in handling corruption. Through extensive experimental evaluation using multiple datasets and models, we found a significant connection between the confidence and robustness of a network.

## 1 INTRODUCTION

With the remarkable success of deep learning architectures in nearly every area of computer vision, a plethora of deep neural networks have emerged. However, convolutional neural networks (CNN) are known to be vulnerable to adversarial attacks, where they can be easily fooled by noise and small perturbations in the input data (Goodfellow et al., 2015). The significant issue arises when these models are found vulnerable to natural corruptions similar to artificial adversarial corruptions (Guo et al., 2020; Agarwal et al., 2020b; Hendrycks & Dietterich, 2019a). The seriousness of this vulnerability can be seen from the fact that these corruptions are inherently present in images Agarwal et al. (2020b) without the hassle of artificially generating them. Another serious problem that acts as a barrier to deep learning model deployment in real-world applications relates to inappropriate calibration of their prediction confidence. In practical scenarios, most models reflect overconfidence in their prediction probabilities even when the model predictions are wrong (Lakshminarayanan et al. (2017)). Existing literature believes when training the model that the higher the confidence of a model better its prediction, whether the testing comes from in or out of distribution. Therefore, to keep that in mind, these networks are designed/trained to make confident predictions on any input as high as possible, which we believe is a reason for their vulnerability. The one famous adversary that exploits this concept is an adversarial example that exploits the training strategy of deep neural networks and primarily uses its ingredients such as gradients while generating artificial perturbations. To overcome adversarial examples, adversarial training has been incorporated which aims to increase the confidence of models against adversarial examples. We believe this confidence in a network that aims to map an image to its label might also be a reason for the success of the backdoor attack because the network maps even the corrupted data with the associated label with high confidence (Agarwal et al., 2023a). We believe this confidence or overconfidence might be a prime reason that these models are vulnerable to even unseen noise types and surprisingly this vulnerability is not associated with any form of deep models whether convolution or transformer (Agarwal et al., 2022a; Gu et al., 2023).

Interestingly, the research focusing on improving the robustness of deep networks has not explored the relationship between model calibration and robustness (Zühlke & Kudenko, 2024; Costa et al.,

2024; Goyal et al., 2023; Han et al., 2023). Current research on CNN robustness primarily focuses on two areas: improving the model's robustness to adversarial attacks (Zhang et al. (2023), Peng et al. (2023)) and distinguishing between real and adversarial images (Sen et al. (2023); Agarwal et al. (2023b)). We are not undermining the effort put in developing these defenses such as binary classifiers which are even generalized in handling unseen perturbations (Agarwal et al., 2021; 2020a) and adversarial training (Qian et al., 2022) which re-train the model using the adversarial images. However, we have to think that in both these effective defense cases, several issues involved: (i) training of a separate classifier, (ii) computational cost in generating adversarial examples, and (iii) trade-off between robsutness and clean accuracy. Henceforth, this research aims to tackle several critical bottlenecks in the existing work: a limited exploration of defense against natural corruption, avoiding training extra classifiers or generation of adversarial examples, and no existing study understanding the correlation between confidence and robustness. Through this work, for the first time, we investigate the underlying reasons for vulnerability against natural corruption, focusing on the role of model confidence. We are particularly interested in the contribution of model confidence in the network's sensitivity in handling corrupted images.

For that, extensive experiments are performed using multiple benchmark object recognition datasets namely CIFAR-10 (Alex, 2009) and CIFAR-100 (Alex, 2009) and classification networks such as VGG (Simonyan & Zisserman, 2014) and PreActResNet (He et al., 2016). We have trained the models using stochastic gradient descent and an advanced version of it namely SWAG (Izmailov et al., 2018) to effectively capture the uncertainty within the model. In brief, the primary contributions of this research are:

1. Identifying overconfident predictions as a key factor contributing to reduced robustness in CNN architectures like VGG and ResNet, particularly against adversarial and corrupted inputs.
2. Employing confidence scores and reliability diagrams to systematically analyze and quantify overconfidence in CNN predictions.
3. Utilizing the SWAG method to improve uncertainty estimation in CNNs by fitting a Gaussian distribution over the stochastic gradient descent trajectory, enhancing model calibration.

## 2 RELATED WORK

**Image Corruptions:** Several studies have explored the susceptibility of CNNs to common corruption. (Guo et al., 2020) shows that motion blur, commonly occurring in real-world scenarios, can significantly degrade deep learning model performance. Additionally, Agarwal et al. (2020b) introduces camera-inspired perturbations, simulating noise from natural conditions and camera imperfections to study their impact on model robustness. Similarly, Özdenizci & Legenstein (2023) focuses on addressing environmental noises like snow introduced by adverse weather conditions using diffusion methods. Dodge & Karam (2016), show that CNNs are in particular vulnerable to blur and Gaussian noise. To evaluate the robustness of neural network models, corrupted versions of standard datasets have been widely used, as proposed by Hendrycks & Dietterich (2019b). These datasets introduce various types of noise and distortions, categorized systematically into different classes.

**Imporving Robustness against corruptions:** Image restoration and enhancing model robustness against various corruptions have been the focus of many studies. For instance, Cui et al. (2023) introduces a multi-scale representation to effectively improve image quality by addressing different levels of blur and noise in corrupted images. Dong et al. (2023) focuses on utilizing multi-scale processing to remove motion blur through residual learning and low-pass filters, offering a comprehensive approach to handling complex distortions. In the context of enhancing images with high contrast or brightness, Tian et al. (2023) provides an extensive discussion on various deep learning methods tailored for low-light conditions. Cheng et al. (2024) proposes a novel denoising method using a truncated loss function within a Res2Net architecture. This technique efficiently suppresses non-Gaussian noise, including impulse noise like shot noise, while preserving crucial image details and edges. Furthermore, Zhu et al. (2023) introduces a method that restores images degraded by various weather conditions, such as snow and fog. The approach learns weather-general features common across different adverse weather types as well as weather-specific features unique to

individual conditions, enhancing the model's adaptability to diverse environmental distortions. Additionally, researchers are also exploring whether there is any connection between corruption and adversarial perturbation that can be employed for a universal defense (Agarwal et al., 2022c;b).

**Confidence and Model Calibration:** Calibrating deep neural networks is crucial for creating reliable and robust AI systems, especially in safety-critical applications. Model calibration (Moon et al. (2020)) refers to the alignment between a model's predicted probabilities and the actual likelihood of those predictions being correct (Wang, 2023). In a well-calibrated model, when the model predicts an event, the model is calibrated if, for all samples where the model predicts a class with confidence of 80%, the true accuracy is also 80% (Guo et al. (2017)). Various methods have been developed for model calibration, Silva Filho et al. (2023), discusses various approaches, including post-hoc adjustments, regularization techniques, and metrics for assessing calibration quality. Techniques like Bayesian inference (Blundell et al. (2015)) and ensemble methods (Valdenegro-Toro (2019)) are widely used for improving model calibration by providing better uncertainty estimates. Stochastic Weight Averaging-Gaussian (SWAG) (Maddox et al. (2019)), which models the weight distribution of stochastic gradient descent (SGD) to approximate a Gaussian distribution, offers a more reliable estimate of uncertainty, helping to identify and address overconfidence in predictions.

In addition to these methods, various techniques have been introduced to distinguish between correct and incorrect predictions (Naeini et al. (2015)). For evaluating the performance of a model's probabilistic predictions, metrics like Negative Log-Likelihood (NLL) are commonly used. NLL measures the likelihood that the model assigns to the true labels, penalizing incorrect or overconfident predictions. A lower NLL indicates that the model's predicted probabilities align well with the true labels, suggesting not only accuracy but also meaningful confidence scores (Guo et al. (2017)). Together, these methods and metrics play a crucial role in developing models that are both accurate and well-calibrated.

$$\text{NLL} = -\sum_{i=1}^{N} \log P(y_i|x_i, \theta)$$

$P(y_i \mid x_i, \theta)$ is the predicted probability assigned by the model to the true label $y_i$ given the input $x_i$ and model parameters $\theta$.

## 3 ASSERTING MODEL CALIBRATION AND CONFIDENCE

In this section, we describe Stochastic Weight Averaging-Gaussian (SWAG) (Maddox et al., 2019), an extension of Stochastic Gradient Descent (SGD) that addresses its limitations, particularly in uncertainty quantification. While traditional SGD optimizes the neural network by converging to a single set of weights, SWAG takes a different approach. It builds on SGD by collecting multiple weight checkpoints throughout training, averaging them to explore a **broader region** of the loss landscape. SWAG then fits a Gaussian distribution to these collected weights, allowing it to capture the inherent uncertainty in the model's parameters more effectively. We assert this better estimation of uncertainty makes the models highly robust against corruption; however, in the literature, no study exists that understands this phenomenon. Since, natural corruption is a serious concern, understanding whether better calibration can lead to a highly robust model can pave the way for developing robust models through effective training rather than developing a new model always whenever new adversarial comes into the picture.

### 3.1 STOCHASTIC GRADIENT DESCENT

In standard stochastic gradient descent training, the model weights are updated using stochastic gradient descent (SGD). The update rule is given by:

$$\theta_t = \theta_{t-1} + \frac{\eta}{B} \sum_{i=1}^{B} \nabla \log p(y_i|f(x_i; \theta)),$$

where $\theta$ represents the model parameters, $\eta$ is the learning rate, $x_i$ and $y_i$ are the input data and labels, $f(x_i; \theta)$ is the neural network with weights $\theta$, and $B$ is the size of the mini-batch. The term $\nabla \log p(y_i|f(x_i; \theta))$ represents the gradient of the log-likelihood concerning the model parameters.

The loss function typically used in this process is the negative log-likelihood combined with a regularization term:

$$\mathcal{L}(\theta) = -\sum_{i=1}^{B} \log p(y_i | f(x_i; \theta)) + \log p(\theta).$$

Here, the regularizer $\log p(\theta)$ helps prevent overfitting by penalizing certain model parameters. However, this maximum likelihood training approach does not account for uncertainty in the predictions or the parameters $\theta$.

## 3.2 STOCHASTIC WEIGHT AVERAGING GAUSSIAN

Stochastic Weight Averaging (SWA) (Izmailov et al. (2018)) is a method that improves model generalization by averaging the weights of a neural network over several iterations of Stochastic Gradient Descent (SGD). Suppose the weights of the model after epoch $i$ are $\theta_i$. Then, the SWA solution after $T$ epochs is given by:

$$\theta_{\text{SWA}} = \frac{1}{T} \sum_{i=1}^{T} \theta_i,$$

With SWAG (Stochastic Weight Averaging-Gaussian)(Maddox et al. (2019)) a Gaussian is fitted with the SWA mean as the first moment and a low-rank diagonal covariance matrix, thus forming an approximate posterior distribution over model weights. SWAG then estimates the covariance structure around the mean. To capture the uncertainty in the weight space, SWAG uses both a low-rank approximation and a diagonal covariance matrix. The low-rank component models the directions in the parameter space where weights vary the most. The diagonal component accounts for variance along each parameter independently, offering a simpler estimate of uncertainty.

$$\Sigma = \frac{1}{K-1} \sum_{i=1}^{K} (\theta_i - \bar{\theta})(\theta_i - \bar{\theta})^T,$$

where $K$ is the total number of checkpoints, $\theta_i$ are the individual model weights, and $\bar{\theta}$ is the mean of the weights.

This allows SWAG to approximate the posterior distribution over model weights as:

$$p(w \mid \mathcal{D}) \approx \mathcal{N}\left(\mathbf{w}_{\text{SWA}}, \frac{1}{2} \cdot (\Sigma_{\text{diag}} + \Sigma_{\text{low-rank}})\right)$$

Using this Gaussian distribution, sample several weight sets $\mathbf{w}_{\text{SWAG}}^i$. Each sampled weight represents a different version of the model, incorporating the variability captured during training. SWAG can provide well-calibrated uncertainty estimates for neural networks across various settings in computer vision. Notably, it achieves a higher test likelihood compared to other state-of-the-art approaches, such as MC Dropout (Gal & Ghahramani (2015)) and temperature scaling (Guo et al. (2017)).

## 4 EXPERIMENTAL RESULTS AND ANALYSIS

In this section, we first discuss the ingredients needed to perform the experiments such as datasets and CNNs. We have used two benchmark datasets namely CIFAR-10 and CIFAR-100 and two CNNs namely VGG and PreActResNet. We train the PreActResNet-164 model and VGG-16 with Batch Normalization on both datasets for 300 epochs. The initial learning rate is set to 0.01 with a weight decay of 0.0002. Stochastic Weight Averaging (SWA) is introduced at epoch 161 to collect the model weights, using a learning rate of 0.05. We have used the pre-defined training and testing split of datasets to evaluate the confidence of the models. The models are trained using two optimization techniques namely Stochastic Gradient Descent (SGD) and Stochastic Weight Averaging Gaussian (SWAG) to reflect the impact of calibration/confidence on their classification performance. In the end, to analyze the correlation between confidence and robustness, we have used the naturally

Table 1: Effect of calibration on the performance of VGG-16 and PreActResNet-164 using CI-FAR datasets. The results are reported in terms of classification accuracy (%). It shows the better-calibrated model has higher robustness.

| Noise Type | VGG-16 | | | | PreActResNet-164 | | | |
| | CIFAR-10 | | CIFAR-100 | | CIFAR-10 | | CIFAR-100 | |
| | SGD | SWAG | SGD | SWAG | SGD | SWAG | SGD | SWAG |
|---|---|---|---|---|---|---|---|---|
| Clean (No Noise) | 90.53 | **95.62** | 64.00 | **76.87** | 90.27 | **94.59** | 67.79 | **80.37** |
| Brightness | 27.80 | **87.72** | 19.03 | **67.15** | 30.66 | **93.04** | 19.34 | **72.22** |
| Contrast | 11.70 | **85.65** | 5.80 | **65.00** | 11.10 | **88.81** | 3.65 | **65.07** |
| Pixelate | 25.57 | **77.56** | 11.67 | **56.33** | 22.03 | **88.37** | 13.80 | **65.52** |
| Jpeg Compression | 14.18 | **61.40** | 10.10 | **40.20** | 18.08 | **84.42** | 11.02 | **58.26** |
| Snow | 20.52 | **78.92** | 13.94 | **54.99** | 28.56 | **86.65** | 13.37 | **60.62** |
| Frost | 16.50 | **77.93** | 11.10 | **54.86** | 20.66 | **88.10** | 11.48 | **62.54** |
| Fog | 12.04 | **87.18** | 9.21 | **64.40** | 12.50 | **91.23** | 6.07 | **67.37** |
| Gaussian Noise | 17.28 | **50.85** | 5.30 | **33.48** | 26.05 | **79.93** | 8.74 | **49.11** |
| Impulse Noise | 23.16 | **59.68** | 7.54 | **39.50** | 25.35 | **72.44** | 6.14 | **44.10** |
| Speckle Noise | 18.81 | **58.96** | 6.75 | **35.89** | 24.89 | **81.54** | 9.88 | **51.53** |
| Shot Noise | 18.72 | **57.41** | 7.20 | **36.18** | 26.02 | **81.69** | 10.61 | **51.84** |
| Motion Blur | 11.53 | **83.60** | 5.62 | **60.49** | 11.80 | **90.01** | 5.72 | **68.57** |
| Glass Blur | 15.95 | **61.76** | 3.20 | **40.79** | 17.52 | **74.44** | 5.13 | **49.44** |
| Defocus Blur | 12.83 | **86.04** | 9.20 | **63.44** | 14.23 | **90.94** | 8.68 | **69.08** |
| Gaussian Blur | 13.15 | **82.93** | 7.40 | **58.45** | 13.23 | **89.95** | 6.96 | **66.78** |
| Zoom Blur | 10.72 | **84.69** | 6.16 | **61.51** | 11.49 | **91.14** | 5.68 | **69.14** |
| Saturate | 29.38 | **88.04** | 19.17 | **57.87** | 30.79 | **91.56** | 18.93 | **62.42** |
| Spatter | 23.62 | **81.89** | 13.5 | **58.76** | 27.08 | **87.12** | 12.82 | **61.97** |
| Elastic Transform | 11.27 | **79.02** | 9.50 | **55.42** | 14.48 | **88.32** | 8.62 | **64.96** |

corrupted images of the test set of the datasets (Hendrycks & Dietterich, 2019a). The corrupted images of the datasets are taken from the following link[1].

To effectively analyze the observation presented in this paper, we have used several metrics proposed by ((Maddox et al., 2019)) namely (i) **confidence:** is defined as the maximum softmax output value in the model's predictions, representing the model's certainty in its output, (ii) **perfect calibration:** In an ideally calibrated model, the predicted confidence directly aligns with the true accuracy, and (iii) **reliability diagram:** We used the modified reliability diagram as introduced in (Maddox et al., 2019) to effectively visualize how accurately the model's confidence reflects its likelihood of correctness across different types of noise and distortions.

### 4.1 RESULTS AND ANALYSIS

In this section, we present the analysis of the results based on different factors. First, the analysis is based on the different model architectures. Moving further, the analysis is based on the different types of noise and, finally, the analysis based on the different optimization and training methods is presented in detail.

### 4.1.1 ANALYSIS OF MODEL ARCHITECTURES

The choice of model architecture significantly affects the capacity and robustness to corruption. For instance, the PreActResNet outperforms the VGG model in terms of the capacity to classify fine-grained classes. As shown in Table 1, the SGD-trained VGG model yields an accuracy of 64.00% as compared to the 67.79% accuracy obtained by the PreActResNet model on the CIFAR-100 dataset. While for coarse-grained image classification, the performance of SGD-trained models on both datasets yields comparable performance. However, we assert that purely utilizing the softmax score as the confidence score yields poor calibration, which can be visible from the higher

---
[1]https://github.com/hendrycks/robustness?tab=readme-ov-file

performance of the SWAG model compared to the SGD-trained models. Stochastic Weight Averaging Gaussian (SWAG), which fits the Gaussian distribution on the first moment of stochastic gradient descent (SGD) and aims to better learn the true posterior distribution. The SWAG-trained VGG model shows a boost of more than 12.87%; whereas, the SWAG-trained PreActResNet model shows a jump of more than 12% on the CIFAR-100 dataset as compared to the VGG-16 model. It demonstrates the capacity of the SWAG model as compared to the traditionally trained models having stationary SGD distribution. We want to highlight that the advantage of SWAG not only lies in increasing the capacity of the models but also in increasing their robustness against natural corruption. On each dataset and each form of corruption, SWAG trained models are found significantly better than the SGD trained models. On top of that, similar to the higher capacity of PreActResNet, the model is found to be more resilient than the VGG model in each form of corruption. For instance, with SGD, PreActResNet-164 achieves 90.01% accuracy on CIFAR-10 under motion blur, whereas VGG-16BN reaches 83.60%. Similarly, on CIFAR-100, PreActResNet-164 with SWAG achieves 69.08% accuracy under defocus blur, compared to 63.44% for VGG-16BN. These differences, ranging from 5% to 10%, underscore the superiority of deeper architectures like PreActResNet-164 in handling complex noise and perturbations more effectively than simpler models such as VGG-16BN.

### 4.1.2 ANALYSIS OF NOISE TYPE

In broad terms the corruption used in this research can be broadly grouped into the following categories: (i) **Digital Noise:** included variations in brightness and contrast that can result from different lighting conditions, which can significantly impact image clarity and mislead image classifiers (Agarwal et al. (2019)). Additionally, pixelation and JPEG compression introduce unique artifacts that degrade image quality, (ii) **Environmental Noise:** includes factors such as snow, frost, and fog can severely degrade image quality, (iii) **Random Noise:** The Gaussian distribution is one of the most commonly observed phenomena in the real world, and it represents a frequent type of corruption in images captured in uncontrolled environments. Similarly, shot noise, an electronic disturbance arising from the discrete nature of light, often affects images. Apart from that, the robustness is also evaluated against two other popular noises namely speckle and impulse, (iv) **Blur Effects:** Consists of motion blur, glass blur, defocus blur, Gaussian blur, and zoom blue to evaluate the model's robustness to various types of blurring artifacts, and (v) **Geometric Transforms:** consists of saturate, spatter, and elastic transform corruptions.

The above description demonstrates that the different corruptions of different forms of data distribution drift in the images; henceforth, we can simply assert the unique impact of each corruption on the robustness of the models. When we analyze the impact of each group of corruption, we see clear trends depending on the type of noise. When the VGG model is trained using the traditional SGD training method, it is found most sensitive (lowest accuracy) against the images corrupted by the blur category on each dataset. For example, the VGG model which yields 90.53% accuracy on the clean test set of the CIFAR-10 dataset, drops down to 10.72% under the effect of zoom blur. Whereas, its performance on the CIFAR-100 dataset drops down to 3.20% concerning glass blur from 64.00% on clean images. However, SWAG-trained models come to the rescue and improve the performance of the model on each dataset and corruption. For example, zoom blur which is found most effective under SGD trained shows a jump from 10.72% to 84.69% on the CIFAR-10 dataset when the VGG is trained using SWAG method. A similar tens-of-fold jump has been noticed against glass blur on the CIFAR-100 dataset. It is to note here that while the SWAG model is found robust in handling any corruption, it is found less robust in handling noise corruption including Gaussian, impulse, and shot corruptions. The observation is consistent against the PreActResNet model as well which is found least robust against noise corruption even when it is trained on the SWAG method. Despite that, we must not ignore the resiliency SWAG brings concerning any corruption.

On the VGG model, saturate geometric corruption is found least effective followed by brightness corruption grouped under digital corruption. However, on both the corruptions, the SWAG model boosts the classification performance drastically. For example, when brightness noise is applied to CIFAR-10, VGG-16BN accuracy takes a huge impact, dropping to 27.80% with SGD which elevates to 87.72%, which is an incredible 215% improvement. On a similar note, PreActResNet-164 also benefits greatly from SWAG, with its accuracy rising from 30.66% to 93.04% on brightness-corrupted CIFAR-10 images. The detailed results concerning the type of noise is also shown in Table 1. These results show that SWAG helps models maintain stable predictions even when the images are blurred, allowing them to extract meaningful information despite poor image quality.

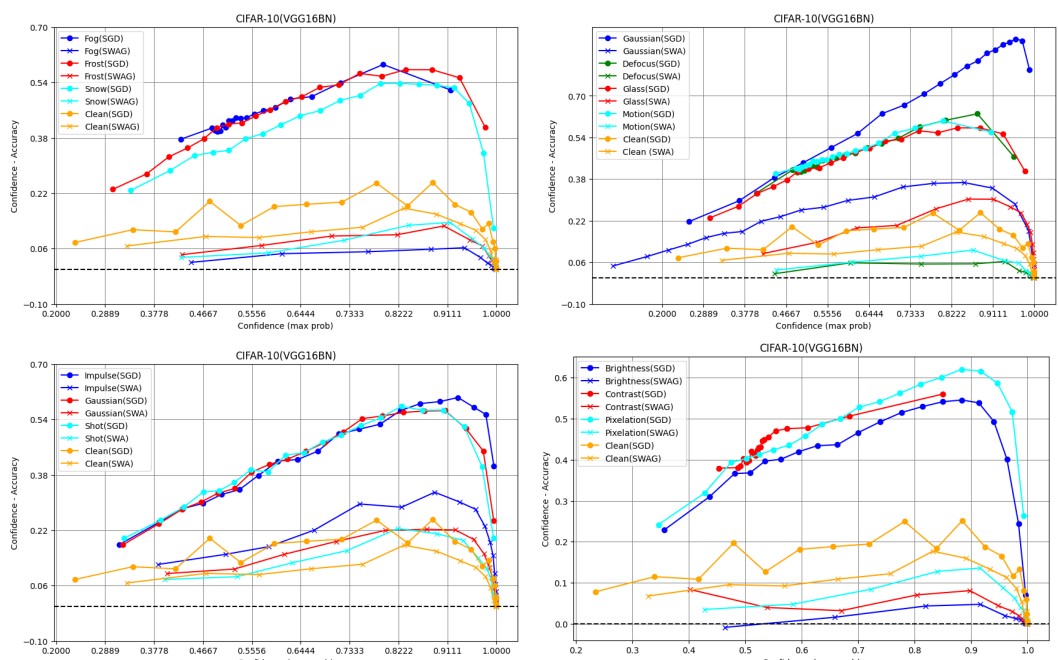

Figure 1: Reliability plots comparing the calibration of different models on CIFAR-10 images corrupted by (i) environmental and blur distortions (top row) and noise and digital corruption (bottom row). The plots are reflected to showcase the calibrated capacity of the VGG model.

### 4.1.3 RELIABILITY ANALYSIS

From Figure 1 on the CIFAR-10 dataset, we observe that the VGG model trained with SGD consistently exhibits overconfident predictions when dealing with noisy or corrupted data. The prediction points are significantly above the optimal line, indicating excessive confidence. This overconfidence is evident in the sharp drop in accuracy at high confidence levels for the SGD-trained models. In contrast, the SWAG-trained models (Maddox et al., 2019)provide more reliable uncertainty estimates, as shown by the smoother curves and higher accuracy across varying confidence levels, particularly under noisy conditions. Although the SGD model also shows overconfidence in the clean dataset, it is far less pronounced than its behavior on data with different types of noise. Its effect can be seen in accuracies in Table 1. In contrast, the predictions made using SWAG are much closer to the optimal line, demonstrating better calibration and improved performance on corrupted data. The reliability curves for the model trained with SWAG are consistently closer to the optimal line, suggesting more reliable and well-calibrated predictions across different noise types. SWAG maintains more calibrated confidence levels across both clean and noisy datasets.

A similar observation from Figures 2 and 3 can be made on the PreActResNet where the predictions made by the SGD model tend to be overconfident when noise is present in the data. This overconfidence is reflected in the model assigning high probabilities to its predictions, even when the input images are corrupted. Such behavior indicates that the SGD-trained model struggles to accurately quantify uncertainty in noisy conditions, potentially leading to incorrect or misleading predictions.

### 4.1.4 EFFECT OF OPTIMIZERS AND TRAINING METHODS

Optimization techniques play a very crucial role in how well models perform, especially when dealing with noisy or corrupted data. For example, on clean CIFAR-10 images, SGD yields an accuracy of 90.53% for VGG-16BN, but SWAG enhances this to 95.62%, representing a substantial gain of 5%. It is to be noted here that all other hyper-parameters are kept fixed when training the model. The distinction between the two strategies becomes more evident with the introduction of noise. For example, in the context of brightness noise, the accuracy of SGD decreases to 27.80%, whereas

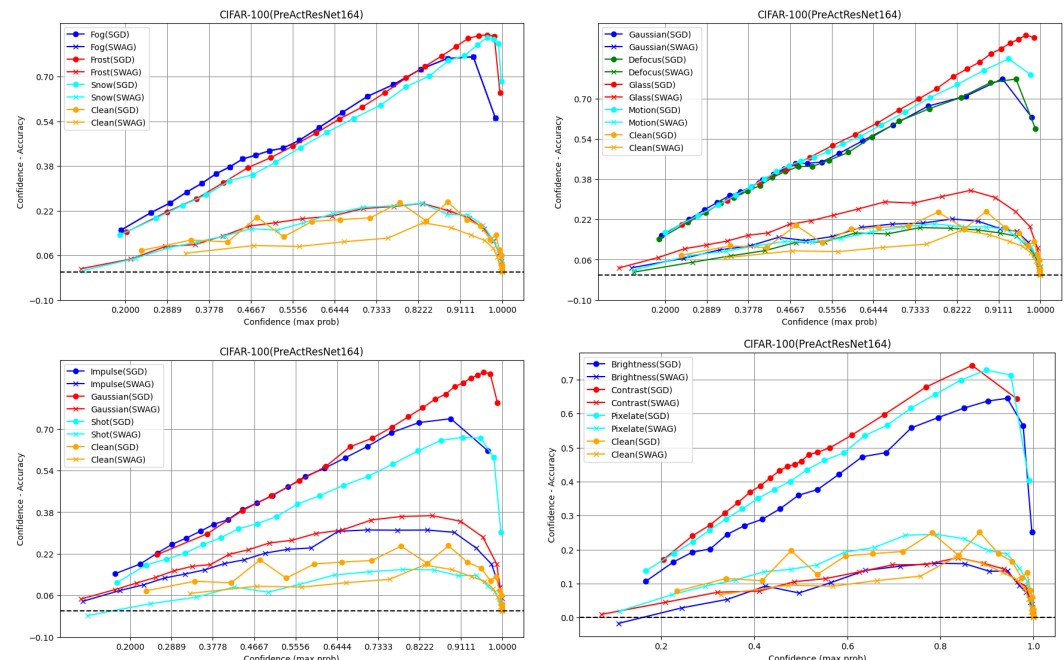

Figure 2: Reliability plots comparing the calibration of different models on CIFAR-100 images corrupted by (i) environmental and blur distortions (top row) and noise and digital corruption (bottom row). The plots are reflected to showcase the calibrated capacity of the PreActResNet model.

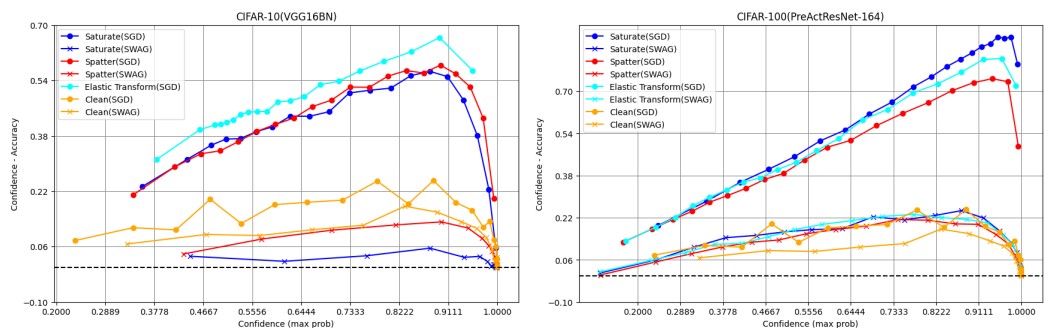

Figure 3: Reliability plots comparing the calibration of VGG and PreActResNet models on CIFAR-10 and CIFAR-100, respectively. The calibration is demonstrated under the influence of geometric corruption.

SWAG suffers a loss of 7.9% on brightness-corrupted images as compared to clean images. This drastic robustness difference is visible across corruptions. In brief, SWAG demonstrates substantial enhancement across noise types, highlighting its proficiency in managing uncertainty and calibration. The observation is not restricted to one dataset or any specific number of classes in the dataset. For example, in the fine-grained CIFAR-100 dataset, the gap between SGD and SWAG is equally prominent. On Gaussian blur images, VGG-16BN attains merely 7.40% accuracy utilizing SGD, while SWAG enhances it to 58.45%, representing an almost 680% increase in accuracy. Similarly, PreActResNet-164 significantly improves the accuracy to 66.78% when subjected to Gaussian blur when SWAG is used as an optimizer. The substantial improvements indicate that SWAG markedly increases the robustness of convolutional neural networks (CNNs) indicating that SWAG is not only improving calibration but also enabling CNNs to generalize better in complex tasks, making it a valuable approach for robustness in adverse conditions.

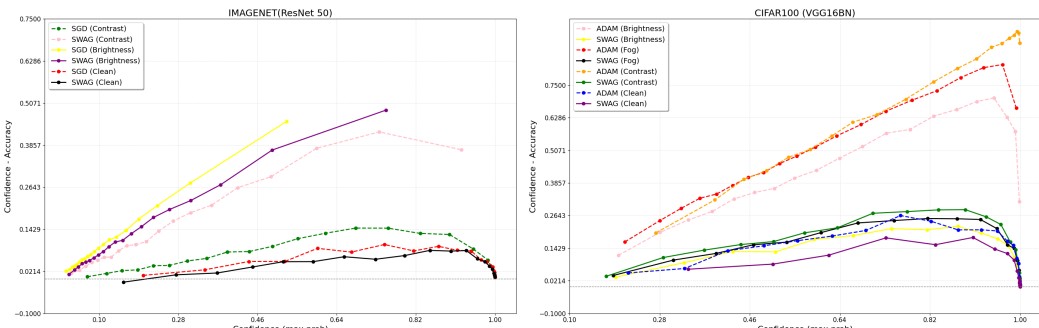

Figure 4: Reliability plots illustrating the impact of corruption on the ImageNet dataset are shown on the left, while the reliability diagram on the right highlights the performance achieved using the Adam optimizer.

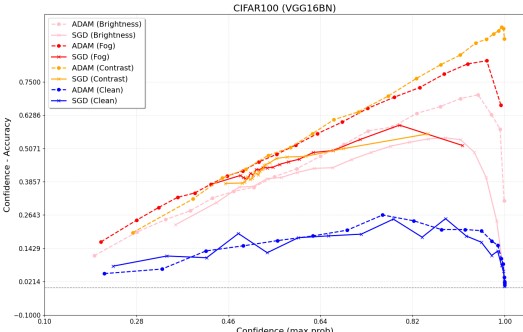

Figure 5: Comparison of SGD and ADAM optimizer for clean data and different environmental noises.

In the case of PreActResNet, trained on CIFAR-10, SGD achieves an accuracy of 90.27%, while SWAG improves this to 94.59%. Similarly, on CIFAR-100, the accuracy for the clean dataset is 67.79% with SGD, but it increases to 80.37% when using SWAG. SWAG provides a more reliable method for handling various forms of corruption, thereby enhancing the performance and robustness of CNNs. Across almost all noise types, both on CIFAR-10 and CIFAR-100 datasets, the models trained with Stochastic Weight Averaging Gaussian (SWAG) show higher accuracy compared to those trained with standard SGD. The most notable improvements with SWAG are seen in challenging noise conditions, such as brightness, contrast, Gaussian blur, and impulse noise, where SWAG significantly enhances model performance. Figure 4 (right) highlights the impact of using the Adam optimizer, different from the one used in Maddox et al. (2019). Even with this modification, the trend remains consistent: models trained with SWAG demonstrate superior calibration compared to their counterparts. Specifically, while Adam-trained models exhibit increased overconfidence in their predictions under different noise conditions, SWAG-trained models maintain predictions closer to the optimal confidence-accuracy line. This observation underscores the robustness of SWAG-trained models, even under varying noise levels and optimizer settings, further validating their effectiveness in handling corrupted dataset. The Figure 5 illustrates the reliability graph, comparing the performance of the SGD and ADAM optimizers under various environmental noise conditions.

### 4.1.5 EFFECT OF LARGER DATASET

The reliability graphs in Figure 4 illustrate model evaluations under various corruption scenarios. On the left, we expand the analysis to include a larger model trained on ImageNet1k, transitioning from CIFAR-100 to assess calibration performance under corruptions such as Contrast, Brightness, and Fog. The findings reveal a consistent trend: models trained with SGD demonstrate increased over-

confidence, even with the larger dataset, while those trained with SWAG exhibit calibration more closely aligned with the ideal confidence-accuracy relationship. Notably, the clean ImageNet accuracy of ResNet-50 improves from 82% with SGD to 91% with SWAG. Under Brightness noise, the accuracy of the SGD-trained model drops significantly to 15%, whereas the SWAG-trained model maintains a much higher accuracy of 47%, highlighting its robustness to such perturbations.

## 5 CONCLUSION AND FUTURE WORK

The tremendous success of deep neural networks has seen a boom in the development of a plethora of architectures; however, interestingly, after the knowledge of their vulnerability against corruption, started a race against developing 'new' robust models. Surprisingly, a few research aims to advance the robustness of existing models. To tackle this issue and understand why the existing models are not robust to natural corruption, we hypothesize this phenomenon from the point of their classification confidence. After conducting a detailed analysis and extensive experimentation, we confirm our hypothesis that overconfidence in predictions leads to vulnerabilities. The reliability diagrams illustrate that, in the presence of natural noise, CNNs trained with standard methods become excessively overconfident in their predictions. Conversely, when training the models using Stochastic Weight Averaging Gaussian, we observed that the confidence scores became more aligned with actual performance, leading to better-calibrated and robust predictions. Thus, for real-world deployment scenarios, it is crucial to consider training with a strategy that can better calibrate the model in its predictions since the world is inherently noisy (Pedraza et al., 2022), (Chen et al., 2023), and every time developing a new robust model leaving a non-robust model behind can lead to a hazardous solution.

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
