# OpenReview forum: "Watch Out!! Your Confidence Might be a Reason for Vulnerability"
_ICLR.cc/2025/Conference — Submitted to ICLR 2025_

### Official Review · Reviewer_qqpm · 2024-10-30

**Soundness:** 2
**Presentation:** 3
**Contribution:** 2
**Rating:** 6
**Confidence:** 3

**Summary:**

This paper investigates the correlation between the confidence of deep neural networks and their vulnerability to natural corruptions. Specifically, the authors leverage the model calibration method SWAG to construct a smoothed model. The parameter of this model is sampled and averaged from the estimated Gaussian distribution of several versions of model parameters recorded during training. The evaluation on the widely used natural corruption benchmark CIFAR-10-C for VGGNet and PreActResNet has shown the robustness of the smoothed model against natural corruptions.

**Strengths:**

- **Novel insight**: this paper is the first to leverage model calibration method to mitigate natural corruptions.
- **Promising experiment results**: the leveraged SWAG method substantially improves the robustness of CNNs against natural corruptions.
- **Well-written paper**: the paper is well-organized and easy to follow.

**Weaknesses:**

- **Limited technical contribution**: the methodology in Section 3 is originally proposed in SWA and SWAG. This paper has not introduced further adjustment or improvement when applying the method to mitigating natural corruptions.
- **Lacking theoretical analysis**: the experiments have shown the effectiveness of SWAG in improving the robustness against corruptions. However, no theoretical analysis is provided to help better understand the source of the robustness.

**Questions:**

- For Figure 1, how is the reliability plot plotted? Specifically, which hyper-parameter is adjusted to control the confidence of the model, and how is it adjusted?
- How does the proposed method perform on larger datasets like ImageNet-C? Can this method survive different set of natural corruptions other than those in CIFAR-10-C, e.g., ImageNet-P, ImageNet-$\bar{C}$ [1]?



[1] On interaction between augmentations and corruptions in natural corruption robustness. NeurIPS 2021.

---

> ### Author Response · Authors · 2024-11-30
> **Response**
>
> **Novelty**
>
> We acknowledge that while SWAG in its raw form is not our contribution; as pointed out by other reviewers, this paper for the first time highlights the key issue: the overconfidence of SGD-trained CNN models and their reduced robustness across various noise and corruption scenarios. It is to be noted here the fact that SGD is one of the most popular optimizers in training any deep models including large models, convolutional models, and transformer architectures. Therefore, understanding its role in making the networks sensitive to corruption itself is a significant contribution. Further, how are there other optimizers that are not explicitly aimed at corruption, can they help in mitigating corruption impact from the angle of network confidence in image classification? As said, this is the first work to explore overconfidence as a key factor underlying the vulnerability of deep neural networks (DNNs). The proposed research aims in that direction and aims to provide a foundational understanding of how overconfidence impacts model robustness, particularly in the presence of natural corruption. **This work lays the benchmark for future research aimed at developing defense algorithms to tackle natural corruptions by knowing the reason why their defense can fail or developing novel calibration techniques with a focus on natural corruptions.**
>
> *Our primary novelty lies in:*
>
> **Highlighting Overconfidence in CNNs:** We systematically analyze and benchmark how CNN architectures, when trained with SGD, exhibit overconfidence in their predictions under corrupted datasets. This issue significantly impacts their robustness and has not been thoroughly explored in the context of corrupted data.
>
> **Reliability Diagrams and Analysis:** We provided detailed reliability diagrams to visualize and quantify the overconfidence of models. Importantly, the generation of these diagrams and the insights derived from them are not tied to SWAG and can be applied independently.
>
> **Benchmark for Corruptions:** Our study serves as a benchmark for understanding and addressing model overconfidence in corrupted datasets, which can guide future research in robustness and calibration.
> Our primary concern and contribution lie in analyzing the behavior of CNNs under corruption and highlighting their limitations in terms of overconfidence and reliability.
>
> **Regarding Larger Datasets and Advanced Models**
>
> **Models:** We acknowledge the initial limitations in the scope of our evaluations. In response to your concerns: Models: We acknowledge the importance of modern architectures, such as Transformers, in benchmarking robustness. While we are actively working to extend our analysis to include such architectures, our current focus on convolutional architectures like VGG and ResNet provides a solid foundation for understanding overconfidence in models. These architectures remain widely used and serve as a meaningful starting point for this research. Preliminary results on Transformer-based architectures show trends similar to those observed in CNNs (e.g., VGG, ResNet), highlighting the generality of our findings across modern architectures. This extension, while ongoing, provides additional support for our contributions.
>
> **Datasets:** We have expanded our analysis to include results on the ImageNet-C dataset, which serves as a more comprehensive and challenging benchmark for assessing model robustness under various corruptions. These experiments reinforce our hypothesis that SGD-trained models, regardless of their architecture, tend to exhibit significant overconfidence when exposed to noisy or corrupted inputs. For example, the clean ImageNet accuracy of ResNet-50 improves from 82% with SGD to 91% with SWAG. When tested under Brightness noise, the accuracy of the SGD-trained model drops drastically to 15%, whereas the SWAG-trained model achieves a considerably higher accuracy of 47%, demonstrating its robustness to such perturbations. These findings highlight the effectiveness of SWAG in mitigating overconfidence and improving robustness in the presence of input corruption.

---

> ### Author Response · Authors · 2024-12-01
> **Response (Part-2): Theoritical Impact**
>
> 1. It is observed when the networks are trained using the SWAG optimization, the loss optima becomes wider which we assert helps the model in achieving robustness to noises. The robustness can be observed from the better performance of the SWAG models on each common corruption significantly improving performance from traditional SGD models. Below, we outline the underlying principles contributing to this robustness:
>
> **Wide Minima in the Loss Landscape:**
> SWAG optimizes the model by converging to wide, flat minima in the loss landscape (Averaging Weights Leads to Wider Optima and Better Generalization, UAI 2018), which are inherently more robust to perturbations. Wide minima ensure that small variations in the input (e.g., noise or corruptions) do not significantly affect the model’s predictions, as the model parameters are less sensitive to such changes.
>
> **Gaussian Weight Averaging:**
> By averaging weights sampled from a Gaussian posterior, SWAG (A Simple Baseline for Bayesian Uncertainty in Deep Learning, NeurIPS 2019) effectively captures a diverse range of plausible solutions. This diversity in the weight space enhances the model's generalization ability and reduces sensitivity to input corruptions, as predictions are averaged across multiple configurations. Additionally, as highlighted in the original SWAG paper, it demonstrates a strong ability to approximate the posterior distribution, enabling thorough exploration of the weight space. This capability contributes significantly to the robustness and adaptability of the model under various challenging scenarios.
>
> **Reduced Overconfidence:** SGD-trained models tend to converge to sharp minima, which are characterized by overconfidence and poor generalization to unseen or corrupted data. SWAG mitigates this issue by regularizing the solution space, resulting in more calibrated predictions that are less likely to be overly confident under corruption.
>
> **Theoretical Connection to Robustness:** The robustness of SWAG can be linked to the geometry of the loss surface: Wide minima are associated with low-curvature regions of the loss surface, which are less affected by noise or distributional shifts. Conversely, sharp minima found by SGD are high-curvature solutions, making them highly sensitive to even minor perturbations in the input.
>
> **Intrinsic Variability in Weight Distributions:** SWAG samples weights from a posterior distribution rather than relying on a single point estimate (as in SGD). This stochasticity introduces resilience by implicitly accounting for uncertainty in the model parameters, making it more robust to corrupted or noisy inputs.
>
>
> 2. **Predictions:** The predictions from the model on the test set are divided into bins based on the confidence score (maximum predicted probability for each prediction). In this case, the data is likely split into 20 bins. For each bin, calculate the average accuracy by checking the actual outcomes of the predictions in that bin. Calculate the average confidence by taking the mean of the confidence scores in that bin. For each bin, the difference between the average confidence and the actual accuracy is plotted on the y-axis, while the x-axis represents the confidence level. The above implementation is aligned to the work proposed in SWAG (A Simple Baseline for Bayesian Uncertainty in Deep Learning, NeurIPS 2019).

---

> ### Comment · Reviewer_qqpm · 2024-12-03
>
> I appreciate the author's effort during the rebuttal period. The response has addressed some of my concerns, e.g., the performance on larger dataset ImageNet-C and other technical details. I will raise my score.
>
> However, after reading the response, the technical contribution of this paper remains to be limited somehow. I agree that this paper has pointed out a novel overconfidence phenomenon of classification models, but it applies an existing method as mitigation. In addition, the theoretical analysis is more conceptual rather than strict.
>
> I hope the authors can further modify the manuscript as stated.

---

> ### Author Response · Authors · 2024-12-03
> **Response**
>
> We want to thank the reviewer for raising the rating and giving us another chance to address the concern.
>
> ---------------------------------------
> **Extension and Novelty:** We have already begun to expand and enhance the methodology to address the limitation of SWAG. One key limitation of SWAG, which we acknowledge, is its unimodal nature. By approximating the posterior distribution as a single Gaussian, SWAG inherently struggles to capture more complex uncertainty landscapes, particularly in cases where the true weight distribution may exhibit multimodality or other non-Gaussian characteristics. Recognizing this constraint, we are actively working on extending the framework to incorporate more flexible and expressive distributions.
>
> Specifically, we are exploring the use of mixtures of Gaussians to model multimodal posterior distributions. This extension allows the framework to represent multiple modes in the uncertainty space, thereby providing a richer understanding of model uncertainty. Sampling from this mixture of Gaussians implicitly generates an ensemble of models, each corresponding to weights from different modes of the loss landscape, which will improve the generalization.
>
> **Our preliminary results using the VGG model on the CIFAR-10 dataset showcase the improvement of at least 5\% across corruptions.** As said, we are actively working in this direction to improve SWAG and are committed to adding these findings to the camera-ready paper.
>
> We believe that these extensions will significantly improve the ability of the framework to address overconfidence and provide a more robust approach to uncertainty modeling.
>
>
> -----------------------
>
> Further, as mentioned earlier and asserted that the proposed approach can provide defense against adversarial attacks as well, we have extensive experiments with different adversarial parameters. The results of two popular adversarial attacks under varying perturbation norms are reported below.
>
> Table 1: Accuracy with different perturbation values under PGD attack using CIFAR-10 dataset and VGG network.
>
> | Epsilon (Max Perturbation) | SWAG    | SGD    |
> |----------------------------|---------|--------|
> | 1/255                      | **75.68**%  | 45.21% |
> | 2/255                      | 72.50%  | 39.50% |
> | 3/255                      | 68.74%  | 34.89% |
> | 4/255                      | 62.08%  | 29.08% |
> | 5/255                      | 57.56%  | 24.37% |
> | 6/255                      | 52.67%  | 19.12% |
> | 7/255                      | 46.87%  | 17.54% |
> | 8/255                      | **41.56**%  | 14.24% |
>
> Table 2: Accuracy with different perturbation values under FGSM attack using CIFAR-10 dataset and VGG network.
>
> | Epsilon (Max Perturbation) | SWAG    | SGD    |
> |----------------------------|---------|--------|
> | 1/255                      | **61.24**%  | 46.78% |
> | 2/255                      | 55.89%  | 41.24% |
> | 3/255                      | 51.24%  | 37.89% |
> | 4/255                      | 47.89%  | 35.29% |
> | 5/255                      | 44.36%  | 32.56% |
> | 6/255                      | 41.15%  | 30.21% |
> | 7/255                      | 38.19%  | 29.43% |
> | 8/255                      | **35.24**%  | 27.17% |
>
> **From the results it can be observed that the SWAG model is not only effective in handling common corruptions but also the adversarial perturbation with a significantly higher margin than SGD.** We believe such universality and extensive analysis can help in building a **universal** defense architecture.
>
> **We hope all these new results address the concerns of the reviewers and look forward to the upgrade to the rating of the paper.**

---

### Official Review · Reviewer_obon · 2024-11-03

**Soundness:** 3
**Presentation:** 4
**Contribution:** 2
**Rating:** 5
**Confidence:** 5

**Summary:**

This paper investigates the vulnerability of deep neural networks (DNNs) when facing natural corruptions (such as noise, blur, etc.) and proposes that the model's confidence could be an important factor contributing to this vulnerability. Experiments demonstrate a significant correlation between a model’s confidence and its robustness in handling corruption. The study primarily focuses on calibrating model confidence and employs the Stochastic Weight Averaging Gaussian (SWAG) method to enhance model robustness.

**Strengths:**

Proposing that high confidence might lead to model vulnerability in naturally corrupted environments is a novel perspective, differing from traditional defense methods.

**Weaknesses:**

1. This paper only conducts experiments on convolutional neural networks (CNNs), lacking tests on other network architectures such as ViT-B/16 or DeiT, to validate the conclusions about confidence and robustness across different model types. This would provide a more comprehensive demonstration of the method's applicability and effectiveness.

2. This paper validates the robustness of the model to natural corruptions and its relationship with confidence using CIFAR-10 and CIFAR-100 datasets. However, the complexity of these datasets is relatively low, making it difficult to fully reflect the model's performance in real-world complex scenarios. It is recommended to conduct further experiments on more challenging datasets such as ImageNet-C or ImageNet-A, which include a broader range of corruptions (e.g., Gaussian noise, motion blur, weather-induced degradation, digital transformations) and better reflect the diversity and complexity of real-world applications.

3. The paper mainly focuses on confidence calibration without an in-depth comparison with other advanced defense methods (such as adversarial training), which may weaken the practical applicability of this approach.

**Questions:**

Although the proposed method addresses natural corruption, its effectiveness against gradient-based adversarial attacks remains unclear. It is recommended that the authors conduct experiments involving FGSM, PGD, and C&W attacks to evaluate the method's performance under adversarial attacks. For example, performance under different noise magnitudes and different numbers of attack iterations could be assessed.

---

> ### Author Response · Authors · 2024-11-30
> **Response**
>
> *Our primary novelty lies in:*
>
> **Highlighting Overconfidence in CNNs:** We systematically analyze and benchmark how CNN architectures, when trained with SGD, exhibit overconfidence in their predictions under corrupted datasets. This issue significantly impacts their robustness and has not been thoroughly explored in the context of corrupted data.
>
> **Reliability Diagrams and Analysis:** We provided detailed reliability diagrams to visualize and quantify the overconfidence of models. Importantly, the generation of these diagrams and the insights derived from them are not tied to SWAG and can be applied independently.
>
> **Benchmark for Corruptions:** Our study serves as a benchmark for understanding and addressing model overconfidence in corrupted datasets, which can guide future research in robustness and calibration.
> Our primary concern and contribution lie in analyzing the behavior of CNNs under corruption and highlighting their limitations in terms of overconfidence and reliability.
>
> **Regarding Larger Datasets and Advanced Models**
>
> **Models:** We acknowledge the initial limitations in the scope of our evaluations. In response to your concerns: Models: We acknowledge the importance of modern architectures, such as Transformers, in benchmarking robustness. While we are actively working to extend our analysis to include such architectures, our current focus on convolutional architectures like VGG and ResNet provides a solid foundation for understanding overconfidence in models. These architectures remain widely used and serve as a meaningful starting point for this research. Preliminary results on Transformer-based architectures show trends similar to those observed in CNNs (e.g., VGG, ResNet), highlighting the generality of our findings across modern architectures. This extension, while ongoing, provides additional support for our contributions.
>
> **Datasets:** We have expanded our analysis to include results on the ImageNet-C dataset, which serves as a more comprehensive and challenging benchmark for assessing model robustness under various corruptions. These experiments reinforce our hypothesis that SGD-trained models, regardless of their architecture, tend to exhibit significant overconfidence when exposed to noisy or corrupted inputs. For example, the clean ImageNet accuracy of ResNet-50 improves from 82% with SGD to 91% with SWAG. When tested under Brightness noise, the accuracy of the SGD-trained model drops drastically to 15%, whereas the SWAG-trained model achieves a considerably higher accuracy of 47%, demonstrating its robustness to such perturbations. These findings highlight the effectiveness of SWAG in mitigating overconfidence and improving robustness in the presence of input corruption.
>
> 2. To further extend our investigation, we conducted additional experiments using PGD attacks on both CIFAR-10 and CIFAR-100 datasets. These experiments revealed consistent trends with our observations on natural corruptions: SGD-trained models exhibit significant overconfidence under adversarial perturbations as well.
>
> **Evaluation of Robustness under FGSM Attacks**
>
> Under the FGSM (Fast Gradient Sign Method) attack, the SGD-trained model achieved an accuracy of 27%, while the SWAG-trained model demonstrated improved robustness with an accuracy of 35%. The hyperparameters for this attack were a maximum perturbation (ϵ\epsilonϵ) of 0.03 with a single-step gradient. The results highlight that SWAG’s convergence to wide minima reduces the model’s sensitivity to small, gradient-based perturbations, which are commonly exploited by FGSM, thereby enhancing robustness compared to the SGD-trained model.
>
> **Evaluation of Robustness under C&W Attacks**
>
> For the Carlini & Wagner (C&W) attack, which employs optimization-based perturbations, the SWAG-trained model achieved an accuracy of 78%, outperforming the SGD-trained model’s accuracy of 71%. The attack was configured with a confidence parameter (ccc) of 10, a learning rate of 0.01, and 1,000 iterations.
>
> **Evaluation of Robustness under PGD Attacks**
>
> Under the iterative PGD (Projected Gradient Descent) attack, the SWAG-trained model showed a significant improvement in robustness, achieving an accuracy of 75% compared to the SGD-trained model’s accuracy of 45%. The hyperparameters used for PGD were a maximum perturbation (ϵ\epsilonϵ) of 0.003 and a step size of 0.008. These results reinforce SWAG’s advantage in achieving robustness, as its wide minima reduce the model’s sensitivity to iterative and accumulated perturbations, unlike SGD, which converges to sharp minima.

---

> > ### Comment · Reviewer_obon · 2024-12-03
> >
> > The maximum perturbation \( epsilon = 0.003 \) is used in PGD attacks, which is less than \( 1/255 \) (a single pixel's intensity level), and is indeed quite small. As such, it may not sufficiently represent realistic adversarial scenarios, and its conclusions could be limited in scope. To provide a more comprehensive evaluation, it would be beneficial to test robustness over a broader range of maximum perturbations, such as \( 1/255 \) to \( 8/255 \). This range aligns better with common adversarial settings and would allow for a deeper understanding of the model's performance under varying levels of perturbation intensity.

---

> ### Author Response · Authors · 2024-12-03
> **Response**
>
> We want to thank the reviewer for raising the rating and giving us another chance to address the concern.
>
> **PGD parameters:** We are now working with different parameters of PGD, while looking at the trend on other perturbations, we are hopeful that can tackled like other perturbations by our model. We will post the findings as soon as we get them, positively before the deadline.
>
> Further to enhance the novelty of our work, we provided the following response:
>
> ---------------------------------------
> **Extension and Novelty:** We have already begun to expand and enhance the methodology to address the limitation of SWAG. One key limitation of SWAG, which we acknowledge, is its unimodal nature. By approximating the posterior distribution as a single Gaussian, SWAG inherently struggles to capture more complex uncertainty landscapes, particularly in cases where the true weight distribution may exhibit multimodality or other non-Gaussian characteristics. Recognizing this constraint, we are actively working on extending the framework to incorporate more flexible and expressive distributions.
>
> Specifically, we are exploring the use of mixtures of Gaussians to model multimodal posterior distributions. This extension allows the framework to represent multiple modes in the uncertainty space, thereby providing a richer understanding of model uncertainty. Sampling from this mixture of Gaussians implicitly generates an ensemble of models, each corresponding to weights from different modes of the loss landscape, which will improve the generalization.
>
> **Our preliminary results using the VGG model on the CIFAR-10 dataset showcase the improvement of at least 5\% across corruptions.** As said, we are actively working in this direction to improve SWAG and are committed to adding these findings to the camera-ready paper.
>
> We believe that these extensions will significantly improve the ability of the framework to address overconfidence and provide a more robust approach to uncertainty modeling.

---

> > ### Author Response · Authors · 2024-12-03
> > **Adversarial Attack Analysis**
> >
> > Thanks again for your comments. As mentioned earlier and asserted that the proposed approach can provide defense against adversarial attacks as well, we have extensive experiments with different adversarial parameters. The results of two popular adversarial attacks under varying perturbation norms are reported below.
> >
> > Table 1: Accuracy with different perturbation values under PGD attack using CIFAR-10 dataset and VGG network.
> >
> > | Epsilon (Max Perturbation) | SWAG    | SGD    |
> > |----------------------------|---------|--------|
> > | 1/255                      | **75.68**%  | 45.21% |
> > | 2/255                      | 72.50%  | 39.50% |
> > | 3/255                      | 68.74%  | 34.89% |
> > | 4/255                      | 62.08%  | 29.08% |
> > | 5/255                      | 57.56%  | 24.37% |
> > | 6/255                      | 52.67%  | 19.12% |
> > | 7/255                      | 46.87%  | 17.54% |
> > | 8/255                      | **41.56**%  | 14.24% |
> >
> > Table 2: Accuracy with different perturbation values under FGSM attack using CIFAR-10 dataset and VGG network.
> >
> > | Epsilon (Max Perturbation) | SWAG    | SGD    |
> > |----------------------------|---------|--------|
> > | 1/255                      | **61.24**%  | 46.78% |
> > | 2/255                      | 55.89%  | 41.24% |
> > | 3/255                      | 51.24%  | 37.89% |
> > | 4/255                      | 47.89%  | 35.29% |
> > | 5/255                      | 44.36%  | 32.56% |
> > | 6/255                      | 41.15%  | 30.21% |
> > | 7/255                      | 38.19%  | 29.43% |
> > | 8/255                      | **35.24**%  | 27.17% |
> >
> > **From the results it can be observed that the SWAG model is not only effective in handling common corruptions but also the adversarial perturbation with a significantly higher margin than SGD.** We belive such universality and extensive analysis can help in building a **universal** defense architecture.
> >
> > We hope all these new results address the concerns of the reviewer and look forward to the upgrade to the rating of the paper.

---

### Official Review · Reviewer_gZH2 · 2024-11-03

**Soundness:** 1
**Presentation:** 2
**Contribution:** 1
**Rating:** 3
**Confidence:** 4

**Summary:**

The paper explores the challenges DNNs face from natural adversarial corruptions, which can undermine their robustness. While past work has focused on detecting and mitigating these corruptions, this study examines whether a model’s confidence may contribute to its vulnerability.

**Strengths:**

1. It explores the vulnerability of DNNs from the perspective of model overconfidence.
2. The article is well-structured and relatively clear in its presentation.

**Weaknesses:**

1. What exactly is the novelty of this paper? SWAG is not your contribution; merely using it to derive some results for analysis does not suffice.
2. The phenomenon of model overconfidence appears to be only a description in your paper. Do you have specific examples or experimental results to substantiate this claim?
3. Is your method limited to CNN architectures? Given the prevalence of transformer-based models, a method solely applicable to CNNs may have limited relevance, and it appears you tested on a very small set of CNN models.
4. Based solely on the text, I cannot appreciate the superiority of your method. Please provide comparative experiments with adversarial training methods, covering dimensions such as effectiveness and cost. Furthermore, does your method apply only to natural corruptions? How would it perform against adversarial samples?
5. The experiments lack depth: (1) In terms of models, this paper tests only on VGG-16 and ResNet, which seems rather limited. Where are the tests on more advanced models? (2) In terms of datasets, you only used CIFAR-10 and CIFAR-100, yet experiments on ImageNet are also necessary.

**Questions:**

Please refer to the weakness.

---

> ### Author Response · Authors · 2024-11-30
> **Response (Part-1)**
>
> We acknowledge that while SWAG in its raw form is not our contribution; as pointed out by other reviewers, this paper for the first time highlights the key issue: the overconfidence of SGD-trained CNN models and their reduced robustness across various noise and corruption scenarios. It is to be noted here the fact that SGD is one of the most popular optimizers in training any deep models including large models, convolutional models, and transformer architectures. Therefore, understanding its role in making the networks sensitive to corruption itself is a significant contribution. Further, how are there other optimizers that are not explicitly aimed at corruption, can they help in mitigating corruption impact from the angle of network confidence in image classification? As said, this is the first work to explore overconfidence as a key factor underlying the vulnerability of deep neural networks (DNNs). The proposed research aims in that direction and aims to provide a foundational understanding of how overconfidence impacts model robustness, particularly in the presence of natural corruption. **This work lays the benchmark for future research aimed at developing defense algorithms to tackle natural corruptions by knowing the reason why their defense can fail or developing novel calibration techniques with a focus on natural corruptions.**
>
> *Our primary novelty lies in:*
>
> **Highlighting Overconfidence in CNNs:** We systematically analyze and benchmark how CNN architectures, when trained with SGD, exhibit overconfidence in their predictions under corrupted datasets. This issue significantly impacts their robustness and has not been thoroughly explored in the context of corrupted data.
>
> **Reliability Diagrams and Analysis:** We provided detailed reliability diagrams to visualize and quantify the overconfidence of models. Importantly, the generation of these diagrams and the insights derived from them are not tied to SWAG and can be applied independently.
>
> **Benchmark for Corruptions:** Our study serves as a benchmark for understanding and addressing model overconfidence in corrupted datasets, which can guide future research in robustness and calibration.
> Our primary concern and contribution lie in analyzing the behavior of CNNs under corruption and highlighting their limitations in terms of overconfidence and reliability.
>
> 2. The issue of model overconfidence is well-documented and supported by various studies. For example:
>
> A. Mitigating Neural Network Overconfidence with Logit Normalization, Proceedings of the 39th International Conference on Machine Learning, PMLR 162:23631-23644, 2022.
>
> B. Rethinking Calibration of Deep Neural Networks: Do Not BeAfraid of Overconfidence, Advances in Neural Information Processing
> Systems 34 (NeurIPS 2021)
>
> C. Confidence-Aware Learning for Deep Neural Networks, ICML'20: Proceedings of the 37th International Conference on Machine Learning
>
> It is to note here that while these studies aim to talk about confidence in a network, they do not aim to understand such confidence or overconfidence is a primary factor of natural corruption sensitivity.
>
> 3. Additionally, we acknowledge the importance of modern architectures, such as Transformers, in benchmarking robustness. While we are actively working to extend our analysis to include such architectures, our preliminary results on Transformer-based architectures show trends similar to those observed in CNNs (e.g., VGG, ResNet), highlighting the generality of our findings across modern architectures. This extension, while ongoing, provides additional support for our contributions.
>
> 4. We acknowledge the importance of conducting comparative experiments with adversarial training methods, as they provide valuable insights into the balance between robustness, effectiveness, and computational cost. However, the primary focus of our study is to highlight the pervasive issue of model overconfidence, particularly in SGD-trained CNN models, when exposed to various types of natural corruption. This key limitation, observed consistently in such models, has been the central point of exploration in our analysis.

---

> > ### Author Response · Authors · 2024-11-30
> > **Response (Part-2): Adversarial Attacks and Beyond**
> >
> > To further extend our investigation, we conducted additional experiments using PGD attacks on both CIFAR-10 and CIFAR-100 datasets. These experiments revealed consistent trends with our observations on natural corruptions: SGD-trained models exhibit significant overconfidence under adversarial perturbations as well.
> >
> > **Evaluation of Robustness under FGSM Attacks**
> >
> > Under the FGSM (Fast Gradient Sign Method) attack, the SGD-trained model achieved an accuracy of 27%, while the SWAG-trained model demonstrated improved robustness with an accuracy of 35%. The hyperparameters for this attack were a maximum perturbation (ϵ\epsilonϵ) of 0.03 with a single-step gradient. The results highlight that SWAG’s convergence to wide minima reduces the model’s sensitivity to small, gradient-based perturbations, which are commonly exploited by FGSM, thereby enhancing robustness compared to the SGD-trained model.
> >
> > **Evaluation of Robustness under C&W Attacks**
> >
> > For the Carlini & Wagner (C&W) attack, which employs optimization-based perturbations, the SWAG-trained model achieved an accuracy of 78%, outperforming the SGD-trained model’s accuracy of 71%. The attack was configured with a confidence parameter (ccc) of 10, a learning rate of 0.01, and 1,000 iterations.
> >
> > **Evaluation of Robustness under PGD Attacks**
> >
> > Under the iterative PGD (Projected Gradient Descent) attack, the SWAG-trained model showed a significant improvement in robustness, achieving an accuracy of 75% compared to the SGD-trained model’s accuracy of 45%. The hyperparameters used for PGD were a maximum perturbation (ϵ\epsilonϵ) of 0.003 and a step size of 0.008. These results reinforce SWAG’s advantage in achieving robustness, as its wide minima reduce the model’s sensitivity to iterative and accumulated perturbations, unlike SGD, which converges to sharp minima.
> >
> > 5. We acknowledge the initial limitations in the scope of our evaluations. In response to your concerns:
> > Models: We acknowledge the importance of modern architectures, such as Transformers, in benchmarking robustness. While we are actively working to extend our analysis to include such architectures, our current focus on convolutional architectures like VGG and ResNet provides a solid foundation for understanding overconfidence in models. These architectures remain widely used and serve as a meaningful starting point for this research. Preliminary results on Transformer-based architectures show trends similar to those observed in CNNs (e.g., VGG, ResNet), highlighting the generality of our findings across modern architectures. This extension, while ongoing, provides additional support for our contributions.
> >
> > **Datasets:** We have expanded our analysis to include results on the ImageNet-C dataset, which serves as a more comprehensive and challenging benchmark for assessing model robustness under various corruptions. These experiments reinforce our hypothesis that SGD-trained models, regardless of their architecture, tend to exhibit significant overconfidence when exposed to noisy or corrupted inputs. For example, the clean ImageNet accuracy of ResNet-50 improves from 82% with SGD to 91% with SWAG. When tested under Brightness noise, the accuracy of the SGD-trained model drops drastically to 15%, whereas the SWAG-trained model achieves a considerably higher accuracy of 47%, demonstrating its robustness to such perturbations. These findings highlight the effectiveness of SWAG in mitigating overconfidence and improving robustness in the presence of input corruption.

---

> > > ### Author Response · Authors · 2024-11-30
> > > **Response: Adversarial Training**
> > >
> > > **Comparison with Adversarial Training (AT):** AT is one of the strongest defense against adversarial perturbation; however, its effectiveness against corruptions are not adequately studied. Although, based on the suggestion of the reviewers, we have performed the comparison of the proposed work with AT. The comparison can be performed on atleast three following perspectives: (i) computational cost, (ii) accuracy on clean images, and (iii) handling of corruptions. The proposed SWAG model is found computationally lighter as compared to AT in terma of training time. For example, the PGD AT model on the CIFAR-10 dataset took approximately 250 minutes; whereas, the computational time of SWAG is 170 minutes on the similar GPU machine. Further, as well known, the AT show significant reduction in clean accuracy. The proposed model not only maintain clean accuracy but even improve it as compared to traditional SGD trained models. The similar effectiveness and strength can be seen in handling corruption, where, the accuracy of the proposed model is atleast 15\% better than the PGD AT model.

---

### Official Review · Reviewer_iz8w · 2024-11-05

**Soundness:** 2
**Presentation:** 2
**Contribution:** 2
**Rating:** 3
**Confidence:** 4

**Summary:**

This paper proposes using Stochastic Weight Averaging Gaussian (SWAG) as a method for calibrating neural networks, aiming to improve their performance and robustness against natural corruptions. The approach leverages SWAG's capacity to model uncertainty and enhance prediction reliability, asserting that better-calibrated confidence scores contribute to robustness in challenging real-world conditions.

**Strengths:**

The paper presents a detailed investigation into the impact of calibration on model robustness, especially under naturally occurring corruptions. By systematically exploring the role of confidence in model predictions, the authors contribute to understanding the relationship between calibration and robustness, reinforcing SWAG’s potential in addressing natural corruption without introducing additional computational burden associated with adversarial training.

**Weaknesses:**

1. Limited Novelty: The paper largely relies on the established SWAG technique without introducing new calibration methods or adaptations specific to the architecture or the problem of natural corruption.

2. Experimental Scope: The experimental evaluations are confined to small datasets (CIFAR-10 and CIFAR-100) and convolutional architectures like VGG and ResNet, lacking analysis on larger-scale datasets and modern architectures like Transformers.

**Questions:**

1. Applicability Across Architectures: The proposed method seems tailored primarily for convolutional neural networks (CNNs). A major gap lies in assessing how well SWAG might generalize to other architectures, such as Transformers, which have become prevalent in vision tasks. Expanding the discussion on generalizability or including Transformers in the experimental setup could enhance the study's relevance and adaptability to current deep learning trends.

2. Novelty in Approach: While the study reinforces known concepts around calibration and robustness, these insights are not novel, particularly within the Bayesian deep learning community, where calibration’s role in improving robustness under adversarial scenarios is well-understood [refA, refB]. This limits the paper's contribution, as it primarily confirms existing knowledge rather than pushing the boundaries with a novel calibration approach. Introducing an innovative calibration technique, or a modified variant of SWAG tailored for robustness, would provide a more substantial contribution.

3. Experimental Limitations: The experiments focus on CIFAR datasets and CNNs, which are both limited in size and scope. A broader evaluation involving larger datasets like ImageNet, and a wider range of architectures, including Transformer-based models, would offer a stronger validation of SWAG's effectiveness. This could also strengthen the paper’s generalizability claims and its relevance for real-world deployment.

References

[refA] Wicker, Matthew, et al. "Bayesian inference with certifiable adversarial robustness." International Conference on Artificial Intelligence and Statistics. PMLR, 2021.

[refB] Stutz, David, Matthias Hein, and Bernt Schiele. "Confidence-calibrated adversarial training: Generalizing to unseen attacks." International Conference on Machine Learning. PMLR, 2020.

---

> ### Author Response · Authors · 2024-11-30
> **Responses**
>
> **Novelty:** While we can say that the paper does not have a technical contribution as per se; however, as pointed out by other reviewers (e.g., Reviewer obon03) this is the first work to explore overconfidence as a key factor underlying the vulnerability of deep neural networks (DNNs). We assert that since the deep networks are fundamentally vulnerable to natural corruption, developing the defenses *without finding out the reason is merely providing a false sense of security*. The proposed research aims in that direction and aims to provide a foundational understanding of how overconfidence impacts model robustness, particularly in the presence of natural corruption. **This work lays the benchmark for future research aimed at developing defense algorithms to tackle natural corruptions by knowing the reason why their defense can fail or developing novel calibration techniques with a focus on natural corruptions.**
>
> Further, while our work builds upon SWAG, we have introduced novel elements and experiments aimed at exploring model confidence, particularly in the context of corrupted images. The traditional SWAG paper has not studied its impact on natural corruption and how the trained using it will behave under natural and adversarial corruption. Furthermore, unlike most existing literature, which primarily focuses on mitigating corruption or distinguishing corrupted images from clean ones, our analysis takes a different perspective. We investigate the underlying reasons for the reduced accuracy of CNN models when exposed to noise, framing this issue through the lens of overconfidence. This novel viewpoint provides fresh insights into the limitations of CNNs under noisy conditions and contributes to a deeper understanding of model robustness and calibration.
>
> **To further address comments, we want to highlight the following contributions:**
>
> **Optimizer Exploration:** Unlike the original SWAG implementation, we experimented with additional optimizers, such as Adam, to evaluate their effect on model performance and confidence under corruption. This extension offers insights into the adaptability of SWAG beyond its original formulation.
>
> **Novelty in Addressing Overconfidence:** To the best of our knowledge, this is the first work to directly investigate and quantify the issue of model overconfidence when exposed to corruption in the input data. Our findings not only highlight the severity of overconfidence in these scenarios but also serve as a benchmark for future research in mitigating this issue.
>
> **Benchmark for Corruption Noise:** By demonstrating the overconfidence phenomenon with corrupted inputs, our paper provides a baseline for evaluating models' susceptibility to noise-induced errors. We believe this contribution is essential for understanding model robustness and calibration.
>
> 2. We have expanded our analysis to address this concern. Specifically, we conducted experiments on the ImageNet-C dataset, which features a diverse range of corruption types, to evaluate the robustness of our approach on a larger-scale dataset. These experiments further validate our findings on the overconfidence issue in models exposed to corrupted inputs.
>
> Additionally, we acknowledge the importance of modern architectures, such as Transformers, in benchmarking robustness. While we are actively working to extend our analysis to include such architectures, our preliminary results on Transformer-based architectures show trends similar to those observed in CNNs (e.g., VGG, ResNet), highlighting the generality of our findings across modern architectures. This extension, while ongoing, provides additional support for our contributions.

---

> > ### Author Response · Authors · 2024-11-30
> > **Transformers**
> >
> > **Results on Transformers:** Our preliminary findings reveals that the transformer models are equally vulnerable to common corruption and overconfidence through SGD is a primary reason. To verify that, have performed experiments with SWAG and observed significant reductions in overconfidence and boost in classification performed on each dataset including ImageNet-C.

---

### Author Response · Authors · 2024-11-30
**Global Responses and Update in Manuscript**

First, we want to thank each reviewer for providing constructive comments and highlighting that this is the first work aiming to provide an explainability in understanding the vulnerability of deep models against natural (common) corruption. It is observed from the literature that several efforts have been started to mitigate the impact of corruption or detect the images as clean or corrupted but no research effort has been made to highlight the reason for its sensitivity in the first place itself. We assert that such interpretability can help build a better and more robust model than later developing one model (extra, and incur additional cost, probably heavy to deploy on computationally limited devices) that can mitigate the impact of corruption or detect them which will further require the purification of corruption.

To further strengthen our findings, we have performed experiments with large-scale ImageNet corruption datasets, adversarial noises, and analysis with different training optimizers.

**Updates in Manuscript:** The modifications can be found on pages 8-9 under section 4.1.4 and in Figures 4 and 5.

**Results on ImageNet-C:**

To address the need for evaluating the proposed method on a broader and more challenging benchmark, we have included experiments on the ImageNet-C dataset. These results align with our observations on CIFAR-10 and CIFAR-100, further demonstrating that SGD-trained models exhibit significant overconfidence when exposed to natural corruption. Additionally, SWAG-trained models consistently outperform SGD in terms of accuracy and robustness under these corruptions, reinforcing the generality of our findings across diverse datasets.

**Analysis with Different Optimizers:**

In response to the suggestion to explore the impact of different optimizers, we extended our experiments to include Adam, alongside SGD. The results reveal that while SGD and ADAM optimizers show varying degrees of overconfidence under natural corruptions, the patterns of improvement achieved by SWAG remain consistent across these optimizers. This highlights the adaptability of SWAG in addressing overconfidence issues, regardless of the underlying optimizer.

**Analysis with different adversarial perturbations are also provided as the responses to individual reviewers who asked for such observations.**

**Results on Transformers** Our preliminary findings reveals that the transformer models are equally vulnerable to common corruption and overconfidence through SGD is a primary reason. To verify that, have performed experiments with SWAG and observed significant reductions in overconfidence and boost in classification performed on each dataset including ImageNet-C.

**Comparison with Adversarial Training (AT):** AT is one of the strongest defense against adversarial perturbation; however, its effectiveness against corruptions are not adequately studied. Although, based on the suggestion of the reviewers, we have performed the comparison of the proposed work with AT. The comparison can be performed on atleast three following perspectives: (i) computational cost, (ii) accuracy on clean images, and (iii) handling of corruptions. The proposed SWAG model is found computationally lighter as compared to AT in terma of training time. For example, the PGD AT model on the CIFAR-10 dataset took approximately 250 minutes; whereas, the computational time of SWAG is 170 minutes on the similar GPU machine. Further, as well known, the AT show significant reduction in clean accuracy. The proposed model not only maintain clean accuracy but even improve it as compared to traditional SGD trained models. The similar effectiveness and strength can be seen in handling corruption, where, the accuracy of the proposed model is atleast 15\% better than the PGD AT model.

**Since now the upload of the revised pdf is not possible, we aim to add all these new findings in the camera ready paper**. We are hopeful that we addressed all the comments of the reviewers and hence looking forward for an updated rating and *acceptance of such a critical work which can pave a way in developing a secure deep learning era of models*. We will be happy to address any remaining comments. Thanks

---

### Author Response · Authors · 2024-12-02
**Awaiting Reviewer's Acknowledgement**

We want to again thank each reviewer for providing the valuable feedback which significantly improved our finding and can pave a way in developing robust system which can be deployed in the real world.

We are actively looking forward to hearing back from the reviewers and meta reviewers in acknowledging our detailed response to each comment raised and would be happy to resolve any remaining comments.

Thanks

---

### Author Response · Authors · 2024-12-03
**Adversarial Attack Analysis and Novel Extensions**

Thanks again for your comments. As mentioned earlier and asserted that the proposed approach can provide defense against adversarial attacks as well, we have extensive experiments with different adversarial parameters. The results of two popular adversarial attacks under varying perturbation norms are reported below.

Table 1: Accuracy with different perturbation values under PGD attack using CIFAR-10 dataset and VGG network.

| Epsilon (Max Perturbation) | SWAG    | SGD    |
|----------------------------|---------|--------|
| 1/255                      | **75.68**%  | 45.21% |
| 2/255                      | 72.50%  | 39.50% |
| 3/255                      | 68.74%  | 34.89% |
| 4/255                      | 62.08%  | 29.08% |
| 5/255                      | 57.56%  | 24.37% |
| 6/255                      | 52.67%  | 19.12% |
| 7/255                      | 46.87%  | 17.54% |
| 8/255                      | **41.56**%  | 14.24% |

Table 2: Accuracy with different perturbation values under FGSM attack using CIFAR-10 dataset and VGG network.

| Epsilon (Max Perturbation) | SWAG    | SGD    |
|----------------------------|---------|--------|
| 1/255                      | **61.24**%  | 46.78% |
| 2/255                      | 55.89%  | 41.24% |
| 3/255                      | 51.24%  | 37.89% |
| 4/255                      | 47.89%  | 35.29% |
| 5/255                      | 44.36%  | 32.56% |
| 6/255                      | 41.15%  | 30.21% |
| 7/255                      | 38.19%  | 29.43% |
| 8/255                      | **35.24**%  | 27.17% |

**From the results it can be observed that the SWAG model is not only effective in handling common corruptions but also the adversarial perturbation with a significantly higher margin than SGD.** We believe such universality and extensive analysis can help in building a **universal** defense architecture.

---------------------------------------
**Extension and Novelty:** We have already begun to expand and enhance the methodology to address the limitation of SWAG. One key limitation of SWAG, which we acknowledge, is its unimodal nature. By approximating the posterior distribution as a single Gaussian, SWAG inherently struggles to capture more complex uncertainty landscapes, particularly in cases where the true weight distribution may exhibit multimodality or other non-Gaussian characteristics. Recognizing this constraint, we are actively working on extending the framework to incorporate more flexible and expressive distributions.

Specifically, we are exploring the use of mixtures of Gaussians to model multimodal posterior distributions. This extension allows the framework to represent multiple modes in the uncertainty space, thereby providing a richer understanding of model uncertainty. Sampling from this mixture of Gaussians implicitly generates an ensemble of models, each corresponding to weights from different modes of the loss landscape, which will improve the generalization.

**Our preliminary results using the VGG model on the CIFAR-10 dataset showcase the improvement of at least 5\% across corruptions.** As said, we are actively working in this direction to improve SWAG and are committed to adding these findings to the camera-ready paper.

We believe that these extensions will significantly improve the ability of the framework to address overconfidence and provide a more robust approach to uncertainty modeling.

**We hope all these new results address the concerns of the reviewers and look forward to the upgrade to the rating of the paper.**

---

> ### Comment · Reviewer_obon · 2024-12-03
>
> PGD attacks typically have higher attack success rates than FGSM because PGD iteratively refines the perturbations, making it more effective at finding adversarial examples. The fact that PGD appears less successful than FGSM in these tables raises questions about the experimental setup.
>
> 1. How many iterations and what step size were used for the PGD attacks? Could the number of iterations be too small, resulting in suboptimal adversarial examples?
>
> 2. Is the PGD attack implementation correct?

---

> ### Author Response · Authors · 2024-12-03
> **Response**
>
> Yes, PGD has a higher attack success rate than the FGSM which can be seen from lower classification accuracy on the PGD attack as compared to the FGSM attack.
>
> For example, when the FGSM attack applied, the network yield 27.17\% accuracy as compared to the value of 14.24\% under PGD attack. Here the networks are trained with SGD. SWAG is found effective in handling iterative PGD attack better than SGD. **So far, the vulnerability of SWAG against any adversarial attack is not explored to the best of our knowledge, hence, general assumption of PGD yielding higher success might not be true.** We will expand this observation in the camera ready paper.
>
> 2. The implementation is correct as we are using the benchmark library to implement the attack. We are further verifying the implementation but we have already verified a couple of times and found no error.
>
> 3. We are further experimenting with varying iterations and step size. Although, we believe our results shows tremendous clues regarding the effectiveness in handling corruption and adversarial perturbations.
>
> Thanks

---

> ### Author Response · Authors · 2024-12-03
> **New Adversarial Results**
>
> Table 3: Accuracy with different perturbation values under PGD attack using CIFAR-10 dataset and VGG network. Here the 50 iterations are used to perform the attack.
>
> | **Epsilon (Max Perturbation)** | **SWAG**  | **SGD**  |
> |--------------------------------|-----------|----------|
> | 1/255                          | 57.67%    | 42.21%   |
> | 2/255                          | 51.56%    | 38.92%   |
> | 3/255                          | 47.24%    | 34.78%   |
> | 4/255                          | 43.59%    | 31.37%   |
> | 5/255                          | 41.65%    | 27.16%   |
> | 6/255                          | 38.21%    | 21.19%   |
> | 7/255                          | 34.12%    | 15.24%   |
> | 8/255                          | 26.23%    | 9.64%    |
>
> After increasing the iterations, we still observed the resilience of SWAG compared to SGD in handling PGD attack. We will keep experimenting and add all possible analysis in the camera ready paper.
>
> **While we wish to perform all the experiments quickly, a limited computational power is a concern and hence request reviewer to see the trend which is clear for different variations of attacks.**
>
> **We hope that we have successfully resolved majority of the concerns. Looking forward for the acknowledgement on the interesting observations resulted from this first ever paper.**

---

> ### Author Response · Authors · 2024-12-03
> **New Results Alert**
>
> **PGD with epsilon = 0.03 and step size = 4/255**
>
> 80 iterations: Even with higher iterations, the accuracy of SWAG is 3 times better than that achieved with SGD.
>
>
> 100 iterations: The accuracy of SWAG is close to 7\%, whereas, SGD yields 0.0\%.
>
> ----------
> With epsilon= 8/255, step size = 2/255, and iterations= 100, the SGD model yields 3.6\% and SWAG yields 19.3\%.
>
> **Further, by these new results, it is to note here that the concern (Reviewer obon) of PGD not performing better than FGSM is also resolved (PGD yield lower accuracy even close to zero sometimes)**.
>
> **These new results also suggest advantage of choosing SWAG over SGD and understanding the fact that SGD inherently bring overconfidence to the networks which leads to their vulnerability to corruption and adversarial perturbations.**
>
> Thanks

---

### Meta-Review · Area_Chair_Jfvm · 2024-12-19

**Metareview:**

This paper focused on studying the underlying reason behind the vulnerabilities of deep neural networks to adversarial corruptions. It found that model confidence could be a key factor. Based on the observation, the paper further proposed using Stochastic Weight Averaging Gaussian (SWAG) for DNN calibration. The experiments on multiple datasets prove the effectiveness of the method.

Different from prior work that tried to detect corrupted images or mitigate corruptions, this paper tried to understand the vulnerabilities and developed a method based on the motivation. Drawing the connection between model confidence and corruption robustness is novel and provides new insights. The paper is well-written. However, there are few weaknesses identified by the reviewers. First, the technical contribution is not enough, as the paper simply applies the Stochastic Weight Averaging Gaussian (SWAG) with some necessary modifications. Second, the initial paper lacks in extensive experiments on larger dataset (e.g., ImageNet) and models (Transformers). The authors made an effort to provide more experiments on larger datasets, more attacks, and new architectures. There are still some concerns about the novelty and comprehensiveness.

After thorough discussions, the reviewers reached a consensus that the paper needs further improvements to address the technical contribution. Therefore, the AC considers that the paper falls short of the ICLR acceptance threshold and recommends rejection.

**Additional Comments On Reviewer Discussion:**

The reviewers initially raised several concerns of the paper:

- Reviewer iz8w raised the concerns about limited novelty and experimental scope of the paper. The authors tried to clarify the novelty of the new observation on the connection between model confidence and corruption robustness. They also provided initial experiments on Transformer models.

- Reviewer gZH2 raised the concerns about technical novelty, limited architectures, and lack of experiments in some aspects. The authors tried to address them by providing detailed description on novelty, extending to new architectures and providing new experiments.

- Reviewer obon raised the concerns about limited architectures, limited datasets, and lack of comparisons with advanced defenses. The authors provided more experiments on other architectures and datasets to address the concerns.

- Reviewer qqpm raised the concerns about limited technical contribution and lacking theoretical analysis. The authors tried to clarify the contribution and the reviewer improved the rating to 6.

After author-reviewer discussion and AC-reviewer discussion, the reviewers and AC reached a consensus that the paper has limited technical contribution and some of the experiments are lacking to sufficiently demonstrate the effectiveness of the method. Therefore, AC would recommend rejection.

---

### Decision · Program_Chairs · 2025-01-22

Reject